# Fermentative α-Humulene Production from Homogenized Grass Clippings as a Growth Medium

**DOI:** 10.3390/molecules27248684

**Published:** 2022-12-08

**Authors:** Alexander Langsdorf, Anna-Lena Drommershausen, Marianne Volkmar, Roland Ulber, Dirk Holtmann

**Affiliations:** 1Institute of Bioprocess Engineering and Pharmaceutical Technology, University of Applied Sciences Mittelhessen, Wiesenstrasse 14, D-35390 Giessen, Germany; 2Institute of Bioprocess Engineering, University of Kaiserslautern, Gottlieb-Daimler-Strasse 49, D-67663 Kaiserslautern, Germany

**Keywords:** green waste, *Cupriavidus necator*, α-humulene, terpenoid, biorefinery, bioeconomy

## Abstract

Green waste, e.g., grass clippings, is currently insufficiently recycled and has untapped potential as a valuable resource. Our aim was to use juice from grass clippings as a growth medium for microorganisms. Herein, we demonstrate the production of the sesquiterpene α-humulene with the versatile organism *Cupriavidus necator* pKR-hum on a growth medium from grass clippings. The medium was compared with established media in terms of microbial growth and terpene production. *C. necator* pKR-hum shows a maximum growth rate of 0.43 h^−1^ in the grass medium and 0.50 h^−1^ in a lysogeny broth (LB) medium. With the grass medium, 2 mg/L of α-humulene were produced compared to 10 mg/L with the LB medium. By concentrating the grass medium and using a controlled bioreactor in combination with an optimized in situ product removal, comparable product concentrations could likely be achieved. To the best of our knowledge, this is the first time that juice from grass clippings has been used as a growth medium without any further additives for microbial product synthesis. This use of green waste as a material represents a new bioeconomic utilization option of waste materials and could contribute to improving the economics of grass biorefineries.

## 1. Introduction

Biotransformation offers promising approaches to convert the chemical industry into a greener and more sustainable industry. One approach is using waste or residual products from other industries to produce basic chemicals. Especially given limited fossil resources, biomass is an increasingly attractive renewable resource that often also comes along with a lower CO_2_ footprint. In the case of lignocellulosic biomass, it is advantageous to use waste material, as it does not compete with food for arable land. While the use of many biomass wastes, especially from agriculture, has been widely studied, the use of green waste has not been comprehensively investigated yet. Green waste is defined as garden and park waste, as well as cuttings from roadside greenery, consisting predominantly of grass clippings and other plant biomass with low lignin content. In major cities, large quantities of potential substrates occur, e.g., in Berlin (3.7 million residents) more than 120,000 tons of herbaceous fresh matter are generated annually [1]. The waste material is mostly composted or applied in biogas plants, although these current methods are not particularly profitable. The mostly herbaceous material consists of carbohydrate-rich polymers such as cellulose and hemicellulose, proteins, and minerals [2], all of which could be put to much better use. However, the composition of green waste is extremely heterologous and varies depending on the season and location, among other factors [2]. The strong heterogeneity is the major obstacle to biomass valorization.

Recently, we have already reviewed potential material utilization methods for green waste [2]. The waste material shows particular potential in biotechnology, where it can serve as a feedstock for microorganisms. Up to 60% of the biomass dry weight is carbohydrates, which can be used as a carbon source [3]. Above all, the composition of the lignocellulose of the plants, consisting of cellulose, hemicellulose, and lignin is important, as it determines the amount and type of sugars in the plants. Cellulose and hemicellulose are macromolecules consisting of monosaccharides, while lignin is an aromatic polymer [4]. The composition of lignocellulose in herbaceous biomass can vary widely depending on the plant species and origin [2]. Furthermore, these plants contain up to 18.5% (dry weight) proteins that can be used by microorganisms [2]. Typically, biomass is hydrolyzed to use the resulting monosaccharides as a carbon source. Fermentative production of platform chemicals from herbaceous biomass hydrolysates has already been demonstrated, e.g., the production of succinic acid from *Miscanthus × giganteus* hydrolysate with *Actinobacillus succinogenes* 130Z [5,6] or the production of xylitol from hydrolysates of North American perennial prairie grass [7] or switchgrass [8] with yeasts.

In this work, we use grass clippings as a substrate for the organism *Cupriavidus necator*, which is a Gram-negative facultative chemolithoautotrophic bacterium that is classified as a β-proteobacterium. The organism is known for its versatile metabolism and can utilize a wide range of carbon sources with maximum specific growth rates between 0.14 and 0.46 h^−1^ [9]. While it can use fructose as well as various amino acids as a carbon source, *C. necator* is not able to use glucose [10]. Previous studies reported the use of many complex substrates like jatropha oil [11,12], vegetable oil [13], soybean oil [14], used cooking oil [15,16], sunflower meal [17], beer brewery wastewater [18], fermented food waste liquid [19], digestate liquors [20], by-products from the biodiesel industry [21], waste glycerol [22], rice paddy straw [23], chicory roots [24], olive mill wastewater [25], and wheat bran [26] as a carbon feedstock for *C. necator*. Naturally, the organism has the ability to produce the bioplastic polyhydroxybutyrate (PHB) [15]. However, metabolic engineering enables the production of various products like alkanes and alkenes [27], alcohols [28,29], methyl ketones [30], 2-hydroxyisobutyric acid [31], or cyanophycin [32]. The strain *Cupriavidus necator* pKR-hum used in this work has the property of producing the terpene α-humulene due to the mevalonate pathway and α-humulene synthase from the shampoo ginger plant *Zingiber zerumbet*, which were introduced by genetic engineering [33]. A variety of other terpenes are formed via the same intermediates according to a sort of modular system [34] as shown by Milker et al. for the terpene β-farnesene [35], allowing the strain to serve as a basis for further terpene syntheses. Terpenoids can be isolated from a variety of plants in the form of essential oils [36]. However, valuable isoprenoids are increasingly produced microbiologically due to improvements in bioengineering [37]. As described by Sonntag et al., the extraction of α-humulene from plants as well as the chemical production of α-humulene is associated with many disadvantages and cannot meet the increasing demand, which is why biotechnological production is preferred [38]. Besides *C. necator*, biotechnological production of α-humulene has already been demonstrated with, e.g., *E. coli* [39], *Methylotuvimicrobium alcaliphilum* [40], or *Methylorubrum extorquens* [38]. α-Humulene shows anti-inflammatory properties [41,42,43], antitumor activity [44,45,46,47,48], and antibacterial and antibiofilm properties [49]. Furthermore, α-humulene is a precursor of the highly potent anticancerogenic agent zerumbone [39]. Therefore, it is a promising substance for pharmaceutical or cosmetic applications.

Juice from herbaceous biomass, especially from grass clippings, has been used for fermentation and the production of chemicals on a limited number of occasions before. The idea to use green grass juice or silage juice for supplementing the cultivation medium of *C. necator* first came up in 2005 in order to minimize the costs of the production of PHB. Koller et al. were able to show that an addition of 5% (*v*/*v*) silage juice increased PHB production by *C. necator* compared to the pure minimal medium [50]. However, the addition of green grass juice did not lead to any improvement in productivity. Unlike Koller et al., Boakye-Boaten et al. could confirm that increasing concentration of grass juice results in higher growth levels and higher product concentrations with *Saccharomyces cerevisiae* and *Lactobacillus brevis* [51]. Andersen et al. [52] and Leiß et al. [53] demonstrated the use of press juice from alfalfa as a fermentation medium for lactobacilli after adjustment of sugar levels. In 2015, Cerrone et al. used press juice from ensiled perennial ryegrass as a carbon source for the production of polyhydroxyalkanoates with *Burkholderia sacchari* and *Pseudomonas chlororaphis* [54]. In 2018, a two-step fed-batch process for the production of PHB with *C. necator* was published, in which untreated nutrient-rich grass press juice was used for cell growth and saccharides deriving from lactic acid fermentation of the press cake for PHB accumulation [55].

Herein, a process will be presented, where juice from grass clippings is used as a growth medium to produce α-humulene with *Cupriavidus necator*. Instead of using biomass merely as a source of carbon, we are striving for a holistic use of biomass. Grass clippings, which make up a large portion of green waste, are used as a model substrate. The growth of *C. necator* as well as terpene production in the grass medium are compared to an established lysogeny broth (LB) medium. The experiments are performed in shaking flasks and are monitored regarding optical density (OD), carbon concentration, nitrogen concentration, pH, and α-humulene yield.

## 2. Materials and Methods

### 2.1. Production of Growth Medium from Grass Clippings

The grass clippings for the experiments were taken from a semi-shaded meadow in the city of Giessen, Germany (50°35′24″ N, 8°40′55″ E) during summer and autumn in 2021. If not used immediately after harvest, the grass clippings were stored at −80 °C until use. The growth medium from grass was prepared via two methods. Firstly, the grass juice was made with a juice extractor. For this purpose the grass was washed with ddH_2_O, dried with a towel, and cut into 1 cm long pieces. The grass fragments were run multiple times through the extractor until the remaining grass was dry. The grass juice from the juice extractor was centrifuged at 16,000× *g* for 20 min at room temperature and the supernatant was collected. Since the production with the juice extractor is very labor-intensive and time-consuming, we switched to the production with a blender after the screening studies. For the second method, 100 g of grass clippings were homogenized with 500 mL of ddH_2_O (200 g/L) for 2 min. The mass was then passed through a cotton cloth and wrung out.

If indicated, the grass juice was either sterile-filtered or autoclaved. When the juice was autoclaved, it was centrifuged afterward at 3000× *g* and 4 °C for 30 min to separate any suspended solids produced during autoclaving. To ensure comparability, experiments being compared are conducted using the same batches of grass juice.

### 2.2. Cultivation of Cupriavidus Necator pKR-hum

The organism *Cupriavidus necator* pKR-hum was constructed in a previous work [33]. Precultures were set up in LB medium with 15 μg/mL tetracycline from cryo-stocks of *C. necator* pKR-hum. The LB medium used in this work consisted of 10 g/L tryptone, 5 g/L NaCl, and 5 g/L yeast extract dissolved in ddH_2_O. The precultures were incubated at 30 °C and 180 rpm until the late exponential phase or early stationary phase. All cultivations were performed in 100 mL shake flasks unless otherwise described. To each culture, 15 μg/mL of tetracycline was added and every culture was inoculated to an OD of 0.1 from the LB preculture. All cultivations were incubated at 30 °C and 180 rpm. In the preliminary screening experiments, the grass juice from the juice extractor was diluted with the minimal medium MMasy. The minimal medium MMasy was previously developed by Sydow et al. [56] and consists of 2.895 g/L Na_2_HPO_4_, 2.707 g/L NaH_2_PO_4_·H_2_O, 0.17 g/L K_2_SO_4_, 0.097 g/L CaSO_4_·2H_2_O, 0.8 g/L MgSO_4_·7H_2_O, 0.943 g/L (NH_4_)_2_SO_4_, and trace elements (1:20,000). The trace elements stock was composed of 15 g/L FeSO_4_·7H_2_O, 2.4 g/L MnSO_4_·H_2_O, 2.4 g/L ZnSO_4_·7H_2_O, 0.48 g/L CuSO_4_·5H_2_O, 1.8 g/L Na_2_MoO_4_·2H_2_O, 1.5 g/L Ni_2_SO_4_·6H_2_O, and 0.04 g/L CoSO_4_·7H_2_O in 0.1 M HCl.

The α-humulene production was induced by adding 0.2% (*w*/*v*) l-rhamnose to the cultivation broth at an OD of 0.5 to 0.7. At the same time, 20% (*v*/*v*) n-dodecane was added to the culture for in situ product removal. n-Dodecane was previously described as a suitable organic phase for the removal of α-humulene during the cultivation of the strain *C. necator* pKR-hum [33]. Samples of 1 mL were taken from the media to measure OD, pH, or TOC and N content and were stored at −20 °C until analysis. OD was measured at 600 nm against the respective medium as blank.

For the continuous recording of growth curves, the Cell Growth Quantifier (Scientific Bioprocessing, Inc., Pittsburgh, PA, USA) was used. The Cell Growth Quantifier (CGQ) is a non-invasive online monitoring system for biomass accumulation in shaking flasks by measuring backscatter intensity. The backscatter intensity is converted to OD using a calibration curve. All measurements in the CGQ were performed in 250 mL shake flasks.

### 2.3. Carbon and Nitrogen Analysis

For the analysis of carbon and nitrogen in the medium, the samples from the aqueous medium were sterile-filtered, centrifuged at 16,000× *g* for 3 min, and diluted with ddH_2_O if necessary. Since the analysis of the usable carbon, like the different sugars via HPLC, is difficult, sum parameters like total organic carbon (TOC) or total nitrogen were used to analyze the consumption of carbon or nitrogen. TOC was determined with a Total Organic Carbon Analyzer TOC-L (Shimadzu Corp., Kyoto, Japan). Cuvette tests were used to determine nitrogen content and consumption. Nitrogen was quantified with Laton Total Nitrogen cuvette tests (Hach Lange GmbH, D-40549 Düsseldorf, Germany) and Vario Total Nitrogen HR cuvette tests (Tintometer GmbH, D-44287 Dortmund, Germany). The cuvettes were analyzed in a DR6000 UV-Vis spectrophotometer (Hach Lange GmbH, D-40549 Düsseldorf, Germany) or an XD 7000 (Vis) spectrophotometer (Tintometer GmbH, D-44287 Dortmund, Germany), respectively.

### 2.4. Quantification of α-Humulene

To quantify α-humulene production, 200 µL samples were taken from the n-dodecane phase of the cultivation. Samples are centrifuged at 16,000× *g* for 3 min and stored at −20 °C until analysis. Prior to analysis, the samples were diluted with acetone 1:10 including 50 mg/L zerumbone as an internal standard. A standard curve was made with α-humulene diluted with n-dodecane. Prior to analysis, the standards were diluted 1:10 with acetone including zerumbone like the samples. 1 μL of samples and standards were applied to a GC-MS system with an HP-5MS GC column (Agilent, Santa Clara, CA, USA).

## 3. Results and Discussion

### 3.1. Screening Studies

In order to identify the optimal grass juice (GJ) concentration and to investigate possible inhibitory effects, different concentrations of grass juice from the juice extractor diluted in a minimal medium (MMasy) without a carbon source were used for the cultivation of *C. necator* pKR-hum. The antibiotic tetracycline was added to all cultures. In combination with the minimal medium, the grass juice serves mainly as a carbon source, since the medium contains all other elements/compounds that *C. necator* needs to grow. Analysis of the grass juice from the juice extractor shows a total organic carbon (TOC) content of 20.0 g/L and a total nitrogen (N) content of 1.2 g/L. Other waste materials used as carbon source for cultivation of *C. necator* show similar TOC concentrations with, for example, 19.7 g/L for rice straw hydrolysate [57] or 23.6 g/L for tuna condensate [58]. The grass juice has an advantageous C/N ratio for microbial growth of about 17. However, it must be mentioned that not all of this carbon and nitrogen can be utilized by the organism. The C/N ratio in the grass juice describes the total C and N concentrations and not what is available for the microorganism. The optimal C/N ratio for the growth of *C. necator* depends on the substrates. For example, an optimal C/N ratio of 6 was determined for food waste-based volatile fatty acids [59], but an optimal ratio of 12 or 24 was determined for various vegetable oils [60].

Figure 1a shows the optical density after 77.5 h of cultivation as a function of the grass juice concentration. The very first thing that can be highlighted is that *C. necator* does grow on the minimal medium supplemented with grass juice. The maximum OD of 10.9 was reached with the medium made of 80% GJ. An OD of 6.2 was reached with 60% GJ, 4.4 with 40% GJ, and 2.5 with 20% GJ. No growth could be detected in the culture that was made of 100% minimal medium without a carbon source. The final optical density increases proportionally to the grass juice concentration. This finding is consistent with previously described results [51]. The data points form a linear function with an R^2^ = 0.976. The slope is 0.13 units of OD per percent grass juice. No evidence of growth inhibition could be detected with increasing concentrations of grass juice. It has previously been demonstrated that the addition of grass juice can be growth-promoting. Koller et al. have shown that the addition of 5% green grass juice or silage juice to the growth medium can promote the growth of *C. necator* [50]. In that case, however, the grass juice was used as an additive and not primarily as a carbon source, although silage juice in particular has a high concentration of sugars. In addition to the increasing final OD, it can be observed that the final pH is higher with increasing grass juice concentration. While the pH in the pure minimal medium without growth is 7.4, 40% GJ shows a final pH of 8.0 and 80% GJ shows a final pH of 8.9. The increasing pH is observed in complex media when e.g., amino acids are consumed by the organism as a carbon source when the cells need more carbon than nitrogen. In this process, ammonia is split off from the amino acid, which forms an ammonium ion in the aqueous medium. As a result, the medium becomes alkaline.

To further investigate the growth, *C. necator* pKR-hum was cultivated in shake flasks with 60%, 80%, and 100% GJ from the juice extractor diluted in MMasy and was monitored online for 24 h by using the Cell Growth Quantifier (Scientific Bioprocessing, Inc., Pittsburgh, PA, USA). The resulting growth curves are presented in Figure 1b. The highest OD could be reached with the medium made of 100% GJ. The culture reached an optical density of 11.6 after 24 h of incubation. A maximum growth rate of 0.65 h^−1^ could be achieved during the exponential phase. Therefore, it can be concluded that *C. necator* is able to grow on grass juice without the addition of minimal medium. Similar results were previously shown by Schwarz et al. who used the diluted and neutralized press juice of grass silage as a growth medium for *C. necator* without any further additives [55]. The culture with 80% GJ grew to an OD of 9 with a maximum growth rate of 0.65 h^−1^ and the culture with 60% GJ grew to an OD of 5 with a maximum growth rate of 0.69 h^−1^. The maximum growth rate during the exponential phase is almost equal for all three concentrations, which again indicates that a higher concentration of grass juice in the medium does not inhibit the growth of *C. necator* pKR-hum. Interestingly, the 100% GJ culture showed the longest lag phase, indicating that the cells needed a little longer to adapt to the grass medium after being cultured in the LB medium. From both experiments, it is evident that higher final OD values are obtained with increasing grass juice concentrations.

After using raw grass juice for fermentation, the next step was to investigate two methods for sterilizing the grass medium. For this purpose, sterile filtration and autoclaving were investigated for their feasibility and influence on the growth behavior of the microorganism. While sterile filtration may not affect the ingredients of the grass medium, it is very tedious to perform. However, autoclaving is expected to change the juice and consequently the microbial growth. In the literature, the (ensiled) grass juice was either autoclaved [54], sterile-filtered [50] or not sterilized at all [55]. The amount of sugar in the grass juice varies greatly from sample to sample depending on the time of harvest or grass composition [61]. Autoclaving will probably render most of these sugars unusable due to the Maillard reaction. In addition to sugars, vitamins or trace elements might be affected. We observed that the grass juice turns into a brown cloudy liquid during autoclaving. Hence, possible substrates are no longer available to the organism. In the following experiment, *C. necator* pKR-hum was cultivated in three flasks with 30% raw, 30% sterile-filtered, or 30% autoclaved grass juice from the juice extractor diluted in MMasy for 6 days and was analyzed for the OD. Again, the antibiotic tetracycline was added to all cultures. The resulting growth curves are shown in Figure 2. The first thing to note is that *C. necator* is growing on all three variations of the grass juice. By comparing the growth curves of the different pretreated grass juices, it can be observed that the untreated and sterile-filtered cultures basically follow the same pattern. Both cultures reach a maximum OD of 1.8 after 50 h, experience a peak at 30 h, and are leveling off in a stationary phase to an OD of 1.8 after 48 h. This indicates that the untreated grass juice contains no microorganisms, which can grow in the tetracycline-supplemented medium and therefore influence the growth in the screening studies. This assumption is confirmed by control cultures without inoculation and microscopic controls of the cultures. The culture grown in the autoclaved medium reached a maximum OD of 1.6 after 56 h of incubation. In comparison to the other two cultures, the growth curve flattens after 30 h, resulting in a slightly lower final OD.

The screening studies have shown that the final optical density increases linearly with higher grass juice concentrations. Accordingly, no inhibitory substance appears to be present in grass juice to any relevant extent. Furthermore, autoclaving the grass juice resulted in similar maximum OD as the raw grass juice, although the growth curve is slightly different. Comparable results were obtained with raw and sterile-filtered grass juice. However, sterile filtration of the particulate grass juice is difficult and it cannot be excluded that contaminations are introduced using raw grass juice. Therefore, all experiments hereinafter were performed with autoclaved grass juice. Furthermore, we switched to the much simpler production of grass juice with a blender at the expense of high substrate concentrations (see Section 2.1). Since it was shown that no supplementation with additional salts and trace elements was needed, only water was used in the production of grass juice, resulting in a growth medium consisting only of homogenized grass clippings and water.

### 3.2. Comparison of Growth and α-Humulene Production in Grass Medium and LB Medium

After demonstrating the successful growth of *C. necator* pKR-hum in the grass juice, the next step focused on the question of whether or not it is possible to produce α-humulene from a fully grass juice-based medium. In addition, the grass medium was compared with the established LB medium. For this purpose, experiments with the autoclaved grass medium and the LB medium were conducted. At the time of induction, an organic dodecane phase was added for in situ product removal of α-humulene. During the cultivation, OD, TOC, nitrogen, pH, and α-humulene concentration were analyzed.

Figure 3 shows the OD and the α-humulene concentration in the organic phase throughout cultivation in the LB medium and the grass medium. After 6 h, product formation was induced with L-rhamnose (indicated by the arrow). The first thing that can be observed is the higher final OD of the grass medium in comparison to the LB medium. A maximum OD of over 8 is reached in the grass medium after 48 h. In the LB medium, the maximum OD of 3.9 is already reached after 13 h. Although the preculture was grown in the LB medium, the culture in the grass medium did not show any adaptation difficulties in the form of longer lag phases compared to the main culture in the LB medium. α-Humulene can be detected in the organic phase of the grass medium after 23 h and thus 13 h later than in the LB medium. In the grass medium, 2 mg/L of α-humulene can be detected at 42 h after induction. In comparison, 10 mg/L of α-humulene can be detected in the LB medium after the same time. However, the production of α-humulene is growth-coupled and thus should be higher in the grass medium than in the LB medium according to the OD values. PHB formation cannot affect OD and α-humulene production because the strain used, *C. necator* pKR-hum, is PHB-deficient. For this reason, we examined the cultures more closely with the microscope. We were able to quickly rule out contamination. But, under the microscope, vesicles can be seen in the grass culture which looks like an oil-in-water (O/W) emulsion (see Appendix A). It can therefore be assumed that an emulsion of dodecane forms in the aqueous grass medium. This emulsion causes the aqueous phase to become turbid and subsequently increases the OD. The formation of vesicles was not observed in the LB medium. The growth of *C. necator* pKR-hum in the grass medium without the addition of dodecane is shown in Appendix A. According to the growth curve without dodecane, the true maximum OD by the microorganisms in the grass medium is about 2.3. Additionally, the maximum growth rate resulting from the growth curve in the grass medium without dodecane is 0.43 h^−1^. In comparison, the maximum growth rate in the LB medium is not much higher, with a value of 0.50 h^−1^. Obviously, there are some until now unknown interactions between the components in the media and the organic phase. Some of the product is presumably trapped in the aqueous phase by the dodecane vesicles. Compared to the LB medium, the α-humulene concentration in the grass medium tends to follow a linear course. An explanation could be that the α-humulene is previously collected in the emulsified dodecane vesicles in the aqueous phase. During another experiment with the same setup, an α-humulene concentration of 1.3 mg/L was achieved after 48 h and a maximum OD of 12 suggesting that less α-humulene is found when there is a higher OD due to stronger emulsification. Treatment of samples from the grass medium with dodecane prior to analysis confirms this assumption. Sterile filtration leaves the suspension turbid, although the microorganisms should have separated. After centrifugation, a small organic phase can be detected on the clear aqueous phase. This emulsion forms only in the presence of microorganisms. In abiotic control cultures, OD did not increase due to the addition of dodecane. Thus, while the in situ product removal is suitable for conventional media, it poses a problem in the complex grass medium.

Subsequently, pH as well as TOC and N concentrations of the cultivation in the grass medium and the LB medium were compared. Figure 4 shows the pH throughout the cultivation in the grass medium and the LB medium. In the LB medium, the pH rises continuously from 7.1 to a value of 9.3 over the cultivation period of 48 h. As previously described in the context of the screening studies, the pH in the grass medium also increases to alkaline. Here, the pH starts at 6.4 and rises to 8.9. However, after 13 h, a dip in the curve is noticeable. This dip occurred in all cultivations with grass medium and can also be observed in the cultivation without the addition of dodecane (see Appendix A). The discontinuous course of the pH value results from the complex composition of the grass medium. Overall, the pH curves are similar due to the shift to alkaline.

Finally, TOC and N were analyzed throughout the fermentation. The results are shown in Figure 5. As expected, TOC and N levels decrease with increasing OD in both cultures. Less organic carbon is available in the grass medium compared to the LB medium. At the beginning of cultivation, the TOC concentration of 6.3 g/L in the LB medium decreases to 4.2 g/L after 48 h. Less than half of the TOC concentration is present in the grass medium. It decreases from 2.7 g/L to 2.0 g/L. The lower carbon consumption is reflected in lower OD and lower productivity within the grass medium. The difference in initial TOC concentration may affect microbial growth, although it is a sum parameter and does not reflect available carbon. The results of the nitrogen analysis show about 1.7 g/L of nitrogen in the LB medium and only 0.2 g/L in the grass medium. Thus, the LB medium has about eight times as much nitrogen as the grass medium. However, the concentration decreases by only 0.2 g/L in the LB medium in comparison to 0.1 g/L in the grass medium. Therefore, the high N concentrations present are not required by the microorganism. The initial values result in a C/N ratio of 11 for the grass medium and a little less than 4 for the LB medium. This is an advantage for the grass medium if the absolute amount of carbon and nitrogen is higher. Since higher optical densities were achieved in the screening studies with more concentrated grass juice, the availability of substrates is likely the limiting factor for microbial growth. Boakye-Boaten et al. have shown that juice from *Miscanthus x giganteus* can have 37.8% elemental carbon and thus a higher carbon concentration than juice from lawn grass [51]. In terms of minerals, potassium is predominant, followed by magnesium and calcium. Components such as minerals in the grass juice and their influence on fermentation need to be analyzed in the future. In addition to an optimized production method, the grass juice can be concentrated by evaporation to obtain higher carbon concentrations as shown by Cerrone et al. [54]. The question remains whether the organism can utilize the substrates before the pH is too alkaline.

All in all, it is difficult to compare the holistic use of grass juice as a growth medium with results from other publications, since in those cases the grass juice is used either only as a carbon source or as an additive. As described earlier, only the juice of grass silage was used as a growth medium for the cultivation of *C. necator* [55]. However, silage juice has a strongly different composition than green grass juice [50]. The main carbon sources are lactic and acetic acid and *C. necator* shows a much lower maximum growth rate of 0.06 h^−1^ in comparison to our experiments [55]. Nevertheless, ensiling can be used to store grass clippings and make them available as a resource throughout the year. However, the necessary storage capacity for ensiling must also be considered.

Milker et al. were able to produce 2 g/L of α-humulene with *C. necator* pKR-hum in a fed-batch process with fructose as a carbon source [62]. To achieve similar values with the grass juice as a growth medium, a similar fed-batch process with pH regulation in a bioreactor would have to be developed. Cerrone et al. have demonstrated a fed-batch process with the feeding of ensiled grass press juice with the organisms *Burkholderia sacchari* IPT101 and *Pseudomonas chlororaphis* IMD555 [54]. To design an efficient process with grass juice as the growth medium, different factors should be optimized. First of all, it must be investigated how the heterogeneity of the raw material affects the success or the course of the fermentation. Although “real” and unpurified grass clippings were used in our experiments, it also remains to be investigated how impurities affect growth and productivity. The raw material must at least be cleaned of pollution such as plastic waste. It is also possible that plants contain substances that have an inhibitory effect on the microorganism.

There is also still a great deal of potential for optimization in the biotransformation of green waste. Krieg et al. highlight that the unique physiological flexibility of *C. necator* allows it to utilize a wide variety of carbon and energy sources [33]. However, the use of substances in biomass can be improved. Identifying the substances that the organism can use for growth could help to optimize the process. There are certainly other microorganisms that can use the nutrients of the grass juice even more extensively. *Cupriavidus necator*, for example, is naturally unable to use glucose. Thus, an organism that can use a variety of sugars as a carbon source would be advantageous. Pretreatment of biomass is mainly intended to make the sugars of cellulose and hemicellulose accessible and to delignify the biomass. However, pretreatment can be energy intensive or might use toxic chemicals, and must be judged to be worthwhile to justify undertaking. Consolidated bioprocessing can additionally help to fully utilize the carbohydrates of lignocellulose by combining enzymatic biomass hydrolysis with product generation in one system. One such consolidated bioprocess was demonstrated by Bokinsky et al. who engineered *E. coli* with the ability to utilize cellulose and hemicellulose from pretreated switchgrass to produce biofuels [63]. A process with an organism that combines these characteristics is also likely to be more robust when dealing with heterogeneous raw material. Nevertheless, only fermentation in a bioreactor is expected to result in significantly higher growth, especially due to pH regulation, and thus in increased α-humulene production. Finally, the method for grass juice production is crucial for the concentration and composition of the grass medium. At laboratory scale, we could already determine strong differences between methods. On a larger, industrial scale, a screw press would probably be most effective, as demonstrated by Cerrone et al. [54] or by Schwarz et al. [55].

The use of green waste juice for fermentation in combination with other novel valorization methods in the form of a biorefinery has great potential to make the reuse of green waste significantly more profitable. Due to the high seasonal variability of green waste, it is probably also beneficial to use different methods at different times of the year. Grass juice as a product is more promising in the summer months when grass clippings with high liquid content are abundant, while in the winter months, e.g., electrodes can be created from carbonized green waste where low moisture is beneficial.

## 4. Conclusions

In our study, we were able to show that homogenized grass clippings can be used as media for the microbial production of high-value chemicals. After demonstrating that *C. necator* grows on a mixture of minimal medium with grass juice as a carbon source, we could also show that *C. necator* grows solely on grass juice without any additives. Finally, we were able to demonstrate the production of α-humulene with *Cupriavidus necator* pKR-hum on a 100% green clippings-based medium. As far as we know, this is the first time that juice from grass clippings has been used as a growth medium without further additives for microorganisms to produce chemicals.

This enables a new value-added production of chemicals based on waste materials. Since various products can be produced with *C. necator*, the results shown here have a platform character. Given current challenges such as climate change, increasing depletion of resources, the growing world population, and a decline in usable agricultural land, new ways of utilizing waste as a material must be found. The results shown represent an option for a bioeconomy based on green waste. Furthermore, an improvement of the financial situation of grass-based biorefineries may also become possible. However, only a comprehensive utilization of green waste in the form of such a biorefinery will ensure an optimized valorization of the waste material.

## Figures and Tables

**Figure 1 molecules-27-08684-f001:**
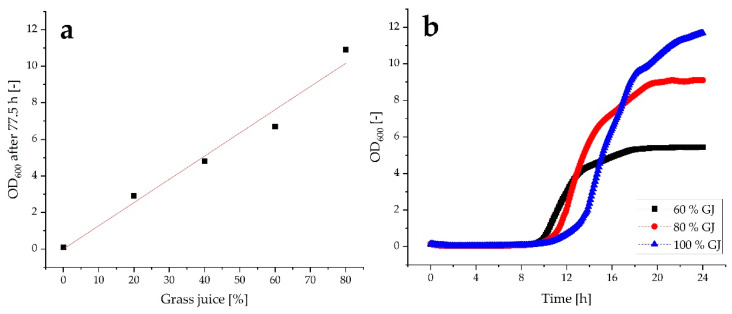
(**a**) Final optical density of *C. necator* pKR-hum culture after 77.5 h with 0%, 20%, 40%, 60%, and 80% of raw grass juice from the juice extractor diluted in MMasy (linear fit with R^2^ = 0.976; *n* = 1); (**b**) growth curves of *C. necator* pKR-hum in 60%, 80%, and 100% raw grass juice (GJ) from the juice extractor diluted in MMasy (recorded with the CGQ; *n* = 1).

**Figure 2 molecules-27-08684-f002:**
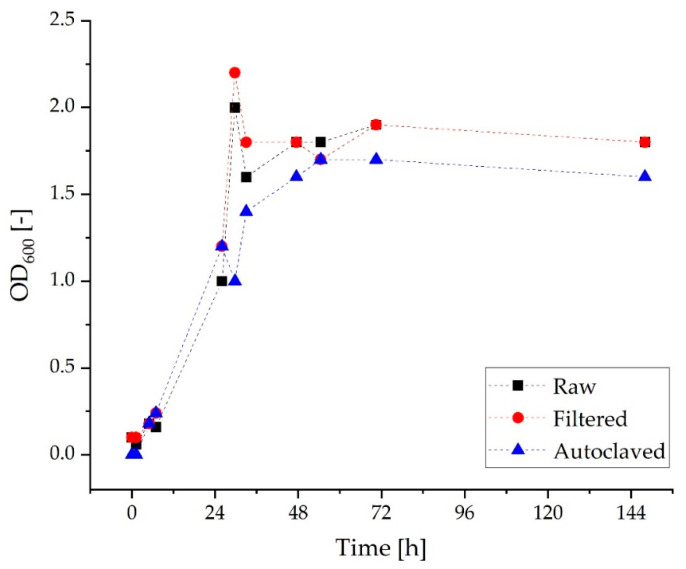
Growth curves of *C. necator* pKR-hum in 30% raw, 30% sterile-filtered, and 30% autoclaved grass juice from the juice extractor diluted in MMasy *(n* = 1, dashed *line* serves as a *guide* to the *eye).*

**Figure 3 molecules-27-08684-f003:**
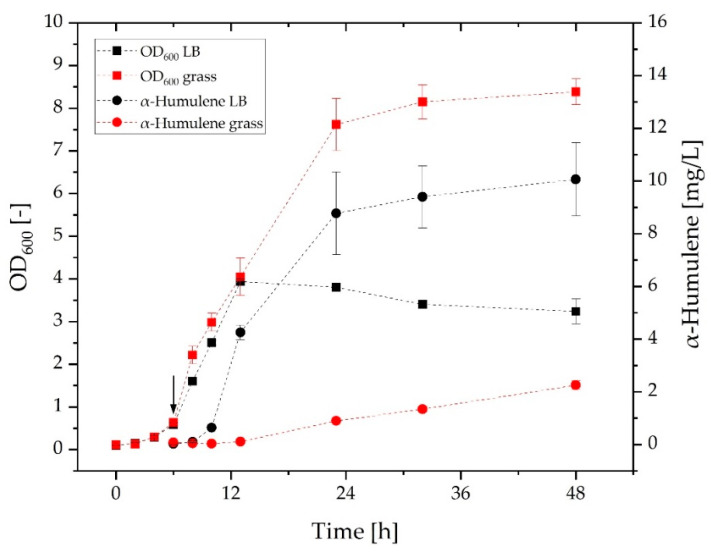
Growth curves of *C. necator* pKR-hum in grass medium and LB medium as well as the respective α-humulene concentration in the organic phase. The induction of α-humulene production after 6 h is indicated by the arrow (*n* = 3, dashed *line* serves as a *guide* to the *eye*).

**Figure 4 molecules-27-08684-f004:**
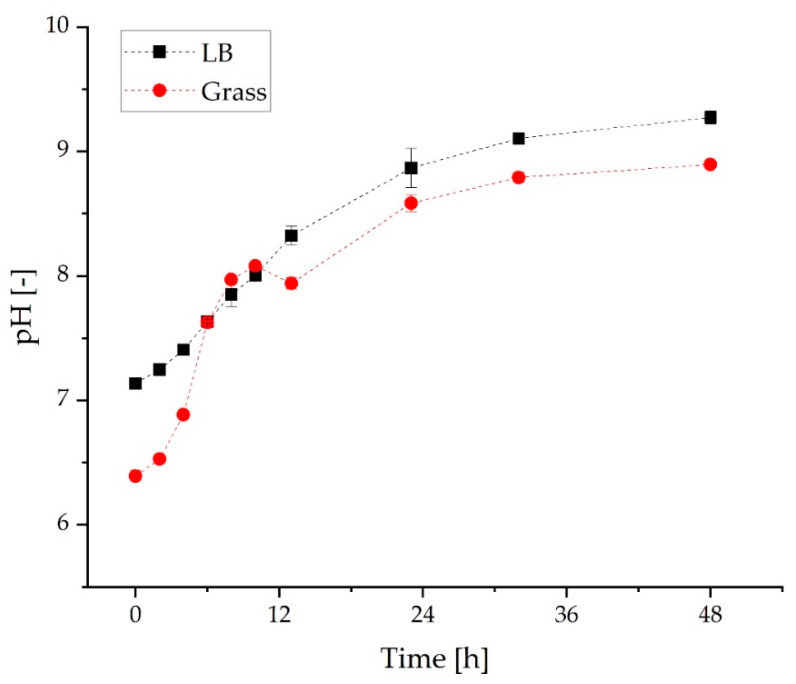
The course of pH of *C. necator* pKR-hum cultivation in LB medium and grass medium (*n* = 3, dashed *line* serves as a *guide* to the *eye*).

**Figure 5 molecules-27-08684-f005:**
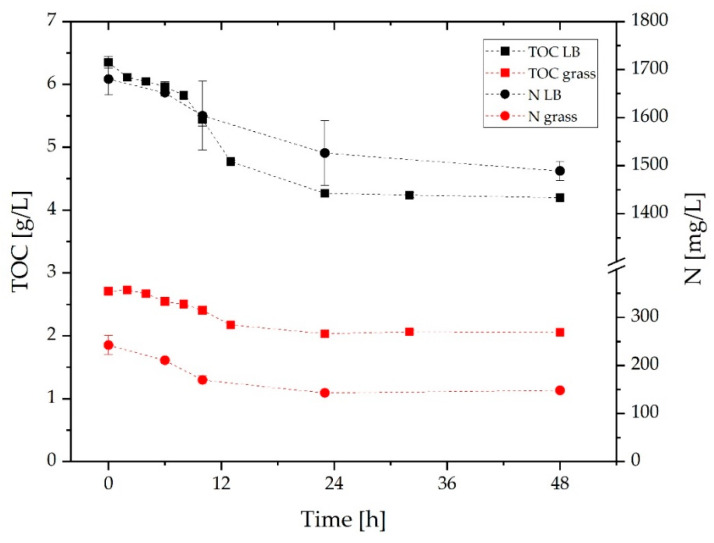
The concentration of TOC and N during the cultivation of *C. necator* pKR-hum in LB medium and grass medium (*n* = 3, dashed *line* serves as a *guide* to the *eye*).

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
