# Peer review of "Fermentative α-Humulene Production from Homogenized Grass Clippings as a Growth Medium"

_molecules, 2022, doi:10.3390/molecules27248684_

Round 1

Reviewer 1 Report

This study demonstrates the production of the sesquiterpene α-humulene with the versatile organism Cupriavidus necator pKR-hum on a growth medium from grass clippings. The use of green waste as a medium has yielded good results and provides a scheme for the utilization of biological waste. This manuscript still needs a major revision.

1. The format of the line chart was not unified. For example, the structure and lines of the line graph in Figure 1, Figure 2 and Figure 3 are not uniform.

2. The format of references in the article needs to be checked, such as the format of reference 8 needs to be unified.

3. There was a lack of comparison and thinking with other people's research results in the discussion.

4. There is too little reference to others' research in the discussion, and the analysis of phenomena is not deep enough.

5. The title of the manuscript is not original enough and does not reflect the highlights of the manuscript. And the material used in the manuscript is grass juice, which is the same as grass clippings.

6. How to define green waste in the abstract.

7. This study also requires a lot of treatment of grass clippings, is the economic effect of this study better? What are the advantages of this study compared to other study.

8. Is it necessary to analyze only total carbon and total nitrogen in the composition analysis of grass juice, but not other components?

9. This study had good results with grass juice as a medium, better than even LB medium. Is there something in it that promotes the growth of Cupriavidus necator pKR-hum?

10. This manuscript mentioned that grass clippings can be used as a “growth medium”. In the article, the waste is only part of the growth medium, which is not accurate.

11. In this study, the use of grass clippings as a growth medium component can promote bacterial growth, but the production of fermentative terpenoid is reduced, and a more rational analysis is needed for the phenomenon related to this..

12. The introduction described in the article needs to correspond to the diagram. For example, the second paragraph of 2.2 should be Figure 3 instead of Figure 4. Please check the full text to avoid such errors.

13.The data for this article requires a significance analysis.

Reviewer 2 Report

Thank you for the opportunity to review this very interesting manuscript.

I have two general questions regarding the design of the experiment:

1. From Figure 5 it can be seen that the mediums compared in Figure 3 had a very different composition at the starting point. It could be assumed that the initial TOC content had an influence on the end result. Please explain why the experiment was not conducted with similar initial TOC levels.

2. Regarding the measurement of the optical density, could the formation of PHB influence the OD and if yes how do you differentiate between cell growth and PHB formation? If no PHB is formed during these experiments, please specify so.

In the introduction or conclusion it would be interesting to discuss the possible applications of alpha-humulene and its economical importance.

Sections order - I would prefer to see the section materials and methods before results and discussion.

More specific comments are found below:

Line 14 define LB before using abbreviation

Lines 43, 49 please specify if % refers to dry weight

Lines 131-133 in order to conclude that grass juice contains all the elements needed for growth, you would need to dilute it with water, not with minimal medium

Figures 1-5 what do n=1 and n=3 mean?

Line 210 did you mean inoculation instead of induction?

Line 352 please specify the year when the grass was harvested

Round 2

Reviewer 1 Report

The manuscript was revised carefully, and the revised manuscript is acceptable.